# Practice Effects, Test–Retest Reliability, and Minimal Detectable Change of the Ruff 2 and 7 Selective Attention Test in Patients with Schizophrenia

**DOI:** 10.3390/ijerph18189440

**Published:** 2021-09-07

**Authors:** Posen Lee, Ping-Chia Li, Chin-Hsuan Liu, Hung-Yu Lin, Chien-Yu Huang, Ching-Lin Hsieh

**Affiliations:** 1Department of Occupational Therapy, College of Medicine, I-Shou University, Kaohsiung 82445, Taiwan; posenlee@isu.edu.tw (P.L.); pingchia@isu.edu.tw (P.-C.L.); 2Department of Occupational Therapy, Kaohsiung Municipal Kai-Syuan Psychiatric Hospital, Kaohsiung 80276, Taiwan; 3Department of Occupational Therapy, College of Medical and Health Science, Asia University, Taichung 41354, Taiwan; otrlin@asia.edu.tw (H.-Y.L.); clhsieh@ntu.edu.tw (C.-L.H.); 4School of Occupational Therapy, College of Medicine, National Taiwan University, Taipei 10051, Taiwan; 5Department of Physical Medicine and Rehabilitation, National Taiwan University Hospital, Taipei 10002, Taiwan

**Keywords:** selective attention, assessment, test–retest reliability, practice effect, schizophrenia, psychometric properties

## Abstract

Background: The Ruff 2 and 7 Selective Attention Test (RSAT) is designed to measure selective attention. It tests automatic detection speed (ADS), automatic detection errors (ADE), automatic detection accuracy (ADA), controlled search speed (CSS), controlled search errors (CSE), and controlled search accuracy (CSA). The purpose of this study was to examine the test–retest reliability, practice effect, and minimum detectable change (MDC) of the RSAT in patients with schizophrenia. Methods: A total of 101 patients with schizophrenia completed the RSAT twice at a 4-week interval. The intra-class correlation coefficient (ICC), paired *t* test, and effect size were used to examine the test–retest reliability and practice effect. The standard error of measurement (SEM) and MDC were calculated. Results: The difference scores between the two assessments were significant in all the indexes. The absolute effect sizes were 0.14 to 0.30. The ICCs of the RSAT ranged from 0.69 to 0.91. The MDC% in the indexes of ADS, ADA, and CSA of the RSAT were <30%. Conclusions: The RSAT is reliable for assessing selective attention in patients with schizophrenia. The RSAT has good to excellent test–retest reliability, a trivial to small practice effect, and indexes of ADS, ADA, and CSA, representing acceptable random measurement error.

## 1. Introduction

Schizophrenia is a common mental disorder causing high disability and is characterized by psychotic symptoms and cognitive impairments. Patients with schizophrenia display impairments of cognitive domains involved in attention, working memory, and executive functioning [1,2,3,4,5,6]. Attention has been frequently reported as the most robust deficit in patients with schizophrenia [7,8,9,10,11]. There are several different types of attention, such as sustained attention, divided attention, and selective attention. Among them, selective attention is especially important for an individual in executing everyday activities [12]. Selective attention is defined as the ability to identify the information most important for an individual from various information inputs and to ignore irrelevant information. This ability is fundamental to an individual in executing higher cognitive functions [13,14]. Such selection is a key aspect of executive function, and it has been reported that patients with schizophrenia have deficits in selective attention. Current evidence suggests that schizophrenia involves significant impairment in the control of the selection of the processes that guide attention to task-relevant inputs. People with schizophrenia may not be able to ignore irrelevant stimuli when attending to stimuli from one designated source [3,15,16]. Furthermore, a cognitive function such as selective attention can predict the functional outcomes of patients with schizophrenia, affect working performance and social function, and even impede treatment and rehabilitation [4]. Therefore, it is essential to identify selective attention in patients with schizophrenia so as to further design proper interventions and to assess selective attention periodically to monitor progress. To achieve these goals, assessment tools that provide precise and sensitive results are needed. However, at this time, the underlying psychometric properties of selective attention assessments in patients with schizophrenia remain obscure.

Reliability concerns the faith that one can have in the data obtained from the use of an instrument. It is the degree to which any measuring tool controls for random error. Reliability, also conceptualized as reproducibility or dependability, is examined to determine whether results of a measurement are consistently the same when the ability of an examinee does not change and the results are free of random measurement error [17,18,19]. Test–retest reliability assessment is crucial in the development of psychometric tools, for it helps to ensure that measurement variation is due to replicable differences between people regardless of time, target behavior, or user profile [20]. On the other hand, random measurement error is often caused by factors that cannot be controlled before the assessment. For example, examinees may make errors due to a lack of concentration, or they may be anxious or nervous because of the unfamiliarity of the tests [21,22]. As random measurement error diminishes, the measurement results move closer to the true scores, and the measurement is more reliable. Therefore, ideally, high test–retest reliability and small random measurement error can ensure the precision of the results in repeated assessments of subjects. Test–retest reliability was the most important test criterion cited by experts for neuropsychological tests used in assessing patients with schizophrenia [23].

Moreover, the practice effect and minimal detectable change (MDC) are also two important factors that affect a measurement’s reliability. A practice effect is an improvement on the task scores because an examinee has repeated exposures to the test materials, not because the examinee has improvements in ability [24]. This situation often occurs in cognitive-related assessments; an examinee may memorize the test questions, leading to better performance [25]. Traditionally, the practice effect has been viewed as a source of error variance. It has been suggested that reducing the practice effect could lead to more accurate interpretations of cognitive outcomes [26].

Standard error of measurement (SEM) and minimal detectable change (MDC) can be used to differentiate between real change and random measurement error. One index of random measurement error that can be used to present the precision of individual results is SEM. SEM is estimated from the standard deviation of a sample of scores at baseline and a test–retest reliability index of the measurement instrument. The minimal threshold beyond random measurement error at certain confidence levels between two assessments is called MDC. MDC is estimated from SEM and a degree of confidence [27]. The MDC is defined as the minimal threshold beyond the random measurement error at certain confidence levels between two assessments. The MDC indicates the minimal magnitude of change that is likely to be real rather than the product of random measurement error and determines if change scores of measurement results are statistically meaningful [28]. It can be used to identify true change in a patient’s performance. The MDC can further be calculated as the MDC percentage (MDC%), which can be used to identify random measurement error. Therefore, it is necessary to examine the reliability, practice effect, MDC, and MDC% of a measurement tool assessing selective attention in patients with schizophrenia.

The Ruff 2 and 7 Selective Attention Test (RSAT) [29] is a selective attention assessment for evaluating sustained attention by utilizing different distractor conditions in the study of voluntary or intentional aspects of attention. The theoretical development rationale of the RSAT is based on automatic and controlled search [29]. The RSAT is suitable for use in patients with schizophrenia because it is easy to access and administer. Moreover, the RSAT identifies the severity level of cognitive impairment concerning daily life functioning [30]. Some studies have suggested that the RSAT might differentiate dysfunctions in the left and right hemispheres [31,32]. The RSAT has also been shown to be reliable in assessing selective attention in various populations with neurological impairments [33,34]. However, the practice effect and the MDC of the RSAT have been examined only in a limited manner, so its utility and the interpretation of the results are restricted.

In clinical practice, it is common to detect cognitive change in patients with schizophrenia by using neurological tests. Since selective attention is one of the important components of cognitive function and the RSAT is an instrument that is often used to detect changes in cognitive function, it is essential to examine the reliability, practice effect and MDC of the RSAT in patients with schizophrenia. Although the test–retest reliability of the RSAT has been reported as adequate to high, with higher coefficients reported for speed than for accuracy scores in other populations [35], little is known about the test–retest reliability, the practice effect or the MDC in patients with schizophrenia, limiting the interpretability and applicability of this measure for research and clinical settings. Thus, the purpose of our study was to examine the test–retest reliability, practice effect, MDC and MDC% of the RSAT in patients with schizophrenia. Accordingly, it is essential to examine the practice effect and to identify the MDC and MDC% to improve the value of the RSAT.

## 2. Materials and Methods

### 2.1. Participants

For this study, a convenience sample was recruited from a psychiatric hospital in Kaohsiung, Taiwan. Inclusion criteria were as follows: (1) diagnosis of schizophrenia according to the Diagnostic and Statistical Manual of Mental Disorders, 5th ed. [36], made by a psychiatrist; (2) stable psychiatric symptoms with a stable dose of antipsychotic medication for at least 3 months to ensure that attention would not change due to changes in symptoms or the effects of drugs, which could support the assumption that the participant’s attention remained stable during the enrollment and study process; (3) scores of ≤3 on the Clinical Global Impression Scale–Severity (CGI-S) [37] for at least 3 months to ensure the illness severity of the participants was stable during the enrollment and study period; (4) scores of >26 on the Chinese version of the Mini-Mental State Examination (C-MMSE) [38], which ensured that patients had sufficient reading or listening comprehension to complete the RSAT; (5) no other diagnoses of psychiatric problems, such as substance abuse, mental retardation, dementia, etc.; (6) age of 20–65 years.

Exclusion criteria were (1) enrollment in another clinical trial; (2) unstable health status, such as an episode of major depression and/or (3) difficulties in recognizing the letters of the English alphabet because of visual or cognitive problems; (4) unstable severity of illness, specifically a change in score of more than 2 on the Clinical Global Impressions Scale–Severity (CGI-S) [37]. A flow chart of the study is shown in Figure 1.

The study protocol was approved by the Institutional Review Board of Kaohsiung Municipal Kai-Syuan Psychiatric Hospital (KSPH-2010-08). Verbal and written information about all experimental details was given to all participants before they provided informed consent. Written informed consent was obtained from the participants prior to experimental data collection. 

### 2.2. Procedure

First, the patients with schizophrenia whom we approached were assessed with the CGI-S and the C-MMSE to ensure that they met our inclusion criteria [37,38,39]. Participants were then assessed with the RSAT by a specially-trained occupational therapist twice, with an interval of 4 weeks. In addition, the CGI-S was administered at enrollment and before retesting of the RSAT to confirm that the symptom severity of the participants had not rapidly changed during the study period. All participants individually received one-on-one RSAT assessments by the same assessor. Such an assessment included instructions, practice, and a formal exam. The total assessment time was about 15 min. During the administration process of the RSAT, the patients were in a quiet room without any distractions affecting their performance. We also collected patients’ demographic characteristics through medical record review.

### 2.3. Measures

#### 2.3.1. Ruff 2 and 7 Selective Attention Test (RSAT)

The RSAT is designed to measure sustained attention and selective attention in individuals aged 16 years to 70 years. This study used the version compiled by the original author, for which the alpha and split-half coefficients for the normative sample are high, suggesting good internal reliability. The RSAT uses a pencil-and-paper form, and the examinee is asked to execute visual search and cancellation tasks for the assessment of selective attention [40,41]. The examinee needs to cancel the digits 2 and 7 in the task. There are 20 trials, each with a 15 s time limit. Each trial contains three lines, in each of which 10 targets are interspersed among 40 non-target items. The first 10 trials are the automatic detection condition. In these 10 trials, letters of the English alphabet are used as distractors, and the examinee needs to cancel digits among the letters. The latter 10 trials are the controlled search condition. In these trials, numbers other than 2 and 7 serve as distractors. In this task, we calculated the raw scores of speed and accuracy in both conditions, namely, automatic detection and controlled search [40]. The speed score is the total number of correct targets identified in 10 trials. The accuracy score is calculated as the number of correct targets divided by the number of correct targets and errors. Therefore, in our study, we calculated 6 indexes of the RSAT: automatic detection speed (ADS), automatic detection errors (ADE), automatic detection accuracy (ADA), controlled search speed (CSS), controlled search errors (CSE), and controlled search accuracy (CSA).

#### 2.3.2. Clinical Global Impression Scale–Severity (CGI-S)

The CGI-S is a measure for examining the severity of psychiatric illness, and it is commonly used in patients with schizophrenia [42,43,44,45,46]. It contains 1 item, which is rated on a 7-point scale ranging from 1 (not at all ill) to 7 (extremely ill), with higher scores indicating greater severity. The CGI-S has good content validity [47]. We used the CGI-S to examine whether the illness severity of the participants was stable at the time of enrollment and during their participation in the study.

#### 2.3.3. Chinese Version of the Mini-Mental State Examination (C-MMSE)

The C-MMSE was used to screen cognitive function in our study. The C-MMSE retains all the terms and components of the original MMSE, which can be categorized into 5 dimensions: time and place orientation, registration, attention and calculation, recall, and language and constructional ability. Unlike the original MMSE, the C-MMSE has 3 additional items, including writing one’s own name and two items on simple calculation [38], so the total score is 33. The cut-point scores for checking cognitive impairment are 26/27 for literate examinees (23/24 in the original MMSE [48,49]). The C-MMSE has good discriminative validity to differentiate adults with different education levels.

### 2.4. Statistical Analysis

We used Statistical Package for Social Science version 22.0 (IBM, Armonk, New York, NY, USA) for data analyses. All *p*-values were set at <0.05 for significant differences, and the *p*-values were two-tailed.

#### 2.4.1. Test–Retest Reliability

The intra-class correlation coefficient (ICC) _(2, 1)_ with 95% confidence intervals (CI) for a two-way random effects model between the first and the second assessments was used to examine the test–retest reliability of the RSAT. ICC values of <0.40, between 0.40 to 0.59, between 0.60 to 0.79, and ≥0.80 respectively indicated poor, fair, good, and excellent reproducibility [50,51,52,53].

#### 2.4.2. Practice Effects

The practice effect was examined by the effect size for the magnitude of the change scores and paired *t*-tests for the significance of the change scores. The effect size was calculated as the mean change scores between the test–retest divided by the standard deviation (SD) of the first assessment [54]. Effect sizes of 0.2, 0.5, and 0.8 respectively indicated small, medium, and large effect sizes [55,56].

#### 2.4.3. Minimal Detectable Change (MDC)

The minimal detectable change (MDC) was calculated based on the ICC value with the following Equations (1) and (2):
(1)$$\mathrm{SEM}={\mathrm{SD}}_{\mathrm{first}\;\mathrm{session}\;\mathrm{of}\;\mathrm{testing}\;\mathrm{scores}}\times (\sqrt{1-\mathrm{ICC}})$$
(2)$$\mathrm{MDC}=1.96\times \sqrt{2}\times \mathrm{SEM}$$

In the above two equations, the SD values were obtained from the testing scores of the first session; the ICC values were obtained from the test–retest reliability; 1.96 is the z-score at the 95% confidence level; and $\sqrt{2}$ was used for the underlying uncertainty during the two repeated assessments.

We also calculated the MDC percentage (MDC%) with Equation (3). The MDC% represents the relative amount of random measurement error. An MDC% below 30% represents acceptable error, and less than 10%, excellent [57,58].
(3)$$\mathrm{MDC}\%=\frac{\mathrm{MDC}}{\mathrm{the}\;\mathrm{mean}\;\mathrm{score}\;\mathrm{of}\;\mathrm{all}\;\mathrm{the}\;\mathrm{trials}}\times 100\%$$

We also used Bland–Altman plots to visually examine the agreement of the RSAT between the two assessments [59]. We assumed that the difference scores followed normal distribution. Therefore, 95% of the difference scores fell between d ± 1.96 × standard deviation. Here, d represents the mean difference scores between the two assessments, and standard deviation is calculated from the deviation of the difference scores of each pair [60,61].

We used Pearson’s *r* to examine the association between the absolute difference scores and the mean scores of each pair of the two repeated assessments. This approach allowed examination of the heteroscedasticity (i.e., a systematic trend). If heteroscedasticity exists, the same MDC value may not be applicable to different functional levels of patients (i.e., selective attention in this study). The absolute value of 0.3 indicated heteroscedasticity in our study [25].

## 3. Results

### 3.1. Demographic and Clinical Characteristics of Participant Subsections

We approached 150 patients with schizophrenia regarding participation in our study. Among them, 35 patients did not meet our inclusion criteria and were excluded. Finally, 115 participants were recruited. Of those 115 participants, 101 completed both sessions of the tests, and 14 participants failed to complete the second session for the following reasons: having no will to continue, having a change in score of more than 2 on the Clinical Global Impressions Scale–Severity (CGI-S) before the second assessment, discharge from the hospital and withdrawal, etc. Therefore, data from 101 patients with schizophrenia were included in the data analyses. During the study process, all patients continued their routine therapeutic activities.

Table 1 shows the characteristics of the participants. The mean age was 44.0 years, and 59.4% of the participants were male. The mean onset age was 23.3 years. The mean duration of psychiatric history was 20.7 years. All participants had more than 6 years of education and exceeded the cut-off scores of the C-MMSE for literate examinees. The mean score of the C-MMSE was 29.9 (SD = 2.5). The CGI-S categories of all patients were largely mild (51.5%) or borderline (37.6%), with some not at all (10.9%). The CGI-S category stayed the same for all participants, indicating that all patients’ abilities were stable during the study process. All participants were receiving maintenance medication (taking anti-psychotic medicine). No significant changes in medication occurred during the study period.

### 3.2. Test–Retest Reliability

Table 2 shows the test–retest reliability of the six indexes of the RSAT. The ICCs for the six indexes of the RSAT between the two assessments ranged from 0.69 to 0.91. 

### 3.3. Practice Effect

The paired t-test showed that all the indexes were significantly different at test and retest (*p* < 0.05). The absolute effect sizes of ADS and ADE were less than 0.2, indicating a trivial effect size. The absolute effect sizes of ADA, CSS, CSE, and CSA ranged from 0.23 to 0.30 (Table 2), indicating a small effect size.

### 3.4. Minimal Detectable Change (MDC) 

Table 2 shows the MDC and the MDC% for the six indexes. The MDC of the six indexes ranged from 8.0 to 43.4. The MDC% of the ADA was less than 10% or within an excellent range, and those of the ADS and CSA were less than 30% or within an acceptable range. On the other hand, the ADE, CSS, and CSE indexes had large MDC%, especially those of the ADE and the CSE, which exceeded 100%.

The Bland–Altman plots of the distributions of the difference scores of the two successive sessions of each participant are presented in Figure 2. The results showed that the difference scores of the ADS, CSS, and CSE had wide ranges with the scores spread out. On the other hand, the difference scores of the ADE, CSA, and ADA had smaller ranges, and the score distributions were concentrated. 

Except for correlation of the CSS index (r = 0.28), the absolute correlations between the difference scores and the mean scores of the two successive sessions were all above 0.3 (r = 0.31 to 0.58).

## 4. Discussion

The aim of the study was to investigate the test–retest reliability, minimal detectable change, and practice effect of the RSAT. Four main results were found in our study. First, the test–retest reliabilities of the six indexes were good to excellent. Second, the practice effects of the six indexes were trivial to small. Third, ADA, ADS, and CSA had acceptable to small MDC%, indicating acceptable random measurement error, while the other three indexes had large MDC%, indicating score instability. Fourth, the MDC values of the six indexes were identified in our study. However, because heteroscedasticity was found in most of the indexes of the RSAT, the MDC values cannot be directly applied. According to our findings, the RSAT is reliable for assessing selective attention in patients with schizophrenia, but its practice effect and random measurement error should be included in the interpretation of its test scores. 

The test–retest reliability of the indexes of the RSAT ranged from 0.69 to 0.91, indicating good to excellent reproducibility. These results were consistent with previous studies investigating test–retest reliability in other populations. Messinis et al. investigated the discriminant validity and test–retest reliability of the RSAT in Greek adults [31]. The results indicated that the test–retest reliability was very high (0.94–0.98) for speed scores (i.e., ADS and CSS) and adequate to high (0.73–0.89) for accuracy scores (i.e., ADE, ADA, CSE, and CSA). Lemay et al. examined the test–retest reliability of several attention and executive function tests, including the RSAT, in middle-aged to elderly subjects [62]. They found that the ICC values of the RSAT ranged from 0.68 to 0.82. Knight et al. examined reliable change indices for the RSAT in older adults and found that the 1-year retest reliability of the RSAT was satisfactory for the speed variables, being in excess of 0.80, but was more modest for the accuracy variables [34]. All speed scores had excellent reliability, with correlation coefficients over 0.80. This is consistent with reliability data reported in [31,34,62] and the test manual. Since the RSAT has been studied in clinical populations, it has been proven to be very sensitive to the severity of illness, and it has also been verified to have good to excellent retest reliability when used in adults. Based on these previous studies and our calculation of ICC values, the RSAT has good to excellent test–retest reliability and is reliable for use in patients with schizophrenia. 

Healthy adults have revealed that a practice effect is common following the repetition of neuropsychological tests. A linear performance increase has also been observed in various tests of attention. Younger adults also demonstrated larger practice effects than those of older participants [34]. Significant practice effects have also been obtained on neuropsychological tests in both short (eight-week) and long (two-year) retest intervals, even in elderly subjects. Knight et al. found that improvements due to practice were consistent across participants of all ages [34], and Messinis et al. found that younger adults demonstrated larger practice effects than those of older participants on speed scores [31]. However, our study found that the practice effect existed in all six indexes; that is, participants had significantly better performance (e.g., better accuracy or higher speed) at the second test. This result is consistent with previous studies investigating the practice effect of the RSAT. Lemay et al. found that all speed scores were subject to a practice effect; performance increases with repetition of a task [62]. The test manual also reported a 10-point increase on both the automatic detection and controlled search speed scores at a six-month retest interval [41]. Knight et al. reported that the test–retest reliability for the speed of visual search was high and that the practice effects during a 12-month period were substantial [34]. Messinis et al. stated that the RSAT was especially sensitive to the practice effect [31]. Due to the practice effect, it is recommended that the practice time before the formal test be increased. Increasing the practice time would familiarize examinees with the test and prevent errors due to unfamiliarity with it. 

This is the first study to identify the MDC% and MDC values of the RSAT in patients with schizophrenia. The results showed that the MDC% in the three indexes of the RSAT were less 30%, indicating acceptable to small random measurement error. The MDC% of the other three indexes ranged from 35.0% to 218.9%, which exceeded our preset criterion of 30%. Since it is highly unlikely that a patient’s selective attention could improve by more than 30% in a short period, the high MDC% values indicate that the scores of these three indexes of the RSAT are unstable. That is, it is difficult to differentiate a patient’s real change from a product of random measurement error. Thus, the real change of a patient with schizophrenia may be over- or underestimated. To reduce the amount of random measurement error of the RSAT, it is suggested that the RSAT be administered two or three times and that the average score of the assessments be used [63,64]. Such an approach can offset the unstable scores caused by random measurement error [65].

Conceptually, the MDC value can be used as the threshold for determining a real change in individual patients in clinical and research settings [65]. For example, a change score exceeding the MDC value can be interpreted as a real change with the corresponding certainty (e.g., 95%). However, the MDC value cannot be directly used for the RSAT for two reasons. First, our study found heteroscedasticity in most indexes of the RSAT. Heteroscedasticity indicates that the amounts of random error vary across patients with different performances (for example, a poorer performance will have larger random measurement error). Accordingly, a fixed value of MDC is not appropriate for patients with diverse levels of performance. The second reason is that the RSAT has a trivial-to-small practice effect, indicating that the scores systematically increase in repeated assessments.

Thus, patients’ change scores are more likely to exceed the MDC value and lead to overestimation. Therefore, when the MDC values are applied, the practice effect should be considered. In such a situation, the MDC%-adjusted MDC can be used for each participant to determine the participant’s real change. Specifically, the MDC%-adjusted MDC value can be calculated with the following equation: mean practice effect ± the first testing score of a participant × MDC%. The MDC%-adjusted MDC value can also be viewed as a reliable change index modified for practice. In clinical settings, clinicians could calculate such values to identify if a patient’s score change is a real improvement, in light of the practice effect and random measurement error. In research settings, researchers, particularly those conducting clinical trials, could report the percentages of patients with change scores larger than such values to identify the effectiveness of interventions.

However, to our knowledge, no study has investigated the test–retest reliability, practice effect, and minimal detectable change of the RSAT in patients with schizophrenia. Compared with other neuropsychological assessments for patients with schizophrenia, the RSAT has good to excellent test–retest reliability, better than those of the Continuous Performance Test (CPT-IP) and Shih–Hsu Test of Attention (SHTA), and possibly equal to those of the Symbol Digit Modalities Test (SDMT), Tablet-based Symbol Digit Modalities Test (T-SDMT), Wisconsin Card Sorting Test (WCST), and Test of Nonverbal Intelligence–Fourth Edition (TONI-4). The practice effect of the RSAT in patients with schizophrenia is similar to those of other neuropsychological assessments. The MDC% ranges from 8.3% to 218.9%, which is a wider range than those of other neuropsychological assessments. According to previous studies comparing middle-aged and elderly people or Greek adults, the practice effect of the RSAT is smaller when applied to patients with schizophrenia, and the test–retest reliability is similar. A comparison of the proposed results with those of related studies is provided in Table 3.

Three limitations should be noted in our study. First, a convenience sample from a psychiatric hospital was adopted in our study. This might have limited the generalizability of our findings. Second, we conducted no validity examinations (e.g., construct validity, known-groups validity, or ecological validity) of the RSAT in this study; such validations are needed to further confirm the psychometric properties of the RSAT in patients with schizophrenia. Third, participants were administered the RSAT only twice for evaluation of the practice effect. Therefore, the practice plateau phase of the RSAT could not be identified. Future studies could increase the number of assessments (e.g., three or four times) to identify the plateau phase of the assessments and better interpret the practice effect of the RSAT. 

## 5. Conclusions

The aim of this study was to investigate the test–retest reliability, practice effect, and minimal detectable change of the RSAT. Our study produced four main findings. First, the test–retest reliabilities of the six indexes were good to excellent. Second, the practice effect existed in all the indexes but was trivial in some. Third, three indexes had acceptable to small MDC%, indicating acceptable random measurement error, while the other three indexes had large MDC%, indicating score instability. Fourth, the MDC values of the six indexes were identified. However, we suggest using a reliable change index modified for practice, due to the heteroscedasticity of the six indexes. Because heteroscedasticity was found in most indexes of the RSAT, the MDC values cannot be directly applied. Based on our findings, some indexes of the RSAT are reliable for assessing selective attention in patients with schizophrenia, but its practice effect and random measurement error should be included in the interpretation of its testing scores. The RSAT has good psychometric properties and quality, and it can be used for repeated neuropsychological assessments, but the results should be interpreted with caution. The practical implications of the study are that the RSAT is recommended for clinical and research applications because it is reliable in patients with schizophrenia. Our sample was a convenience sample of inpatients. In future studies, it will be necessary to expand the scope of sampling.

## Figures and Tables

**Figure 1 ijerph-18-09440-f001:**
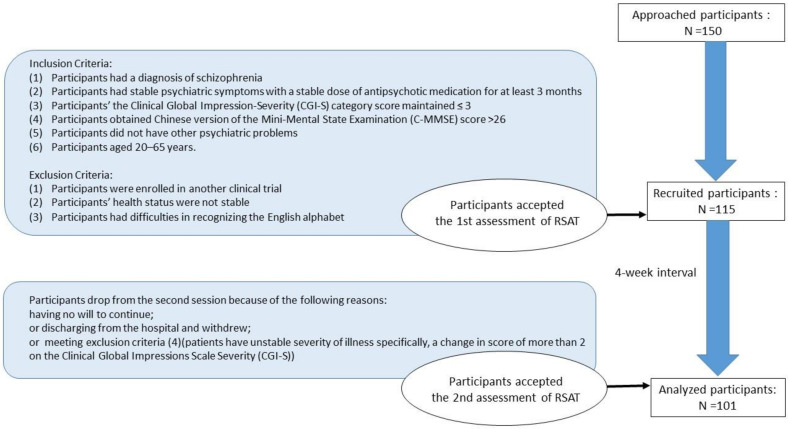
The flowchart of the study.

**Figure 2 ijerph-18-09440-f002:**
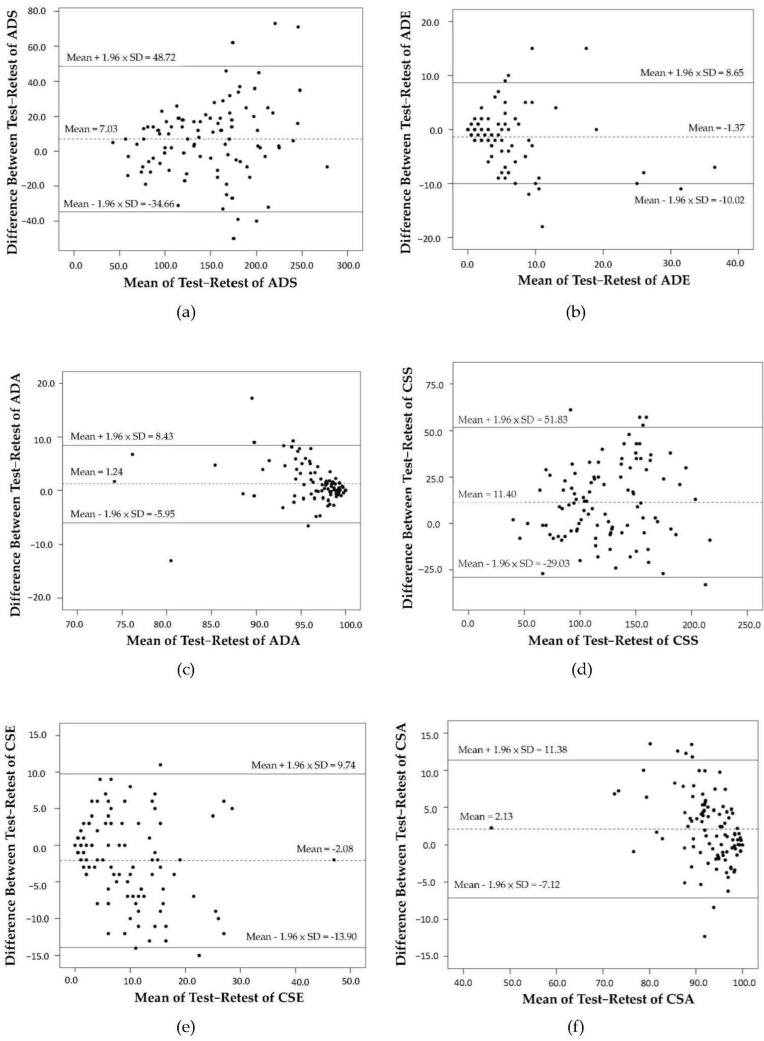
Bland–Altman plots of the differences in scores against the mean scores of the RSAT for (**a**) ADS; (**b**) ADE; (**c**) ADA; (**d**) CSS; (**e**) CSE; (**f**) CSA. The two bold lines define the limits of agreement (mean of difference ± 1.96 SD).

**Table 1 ijerph-18-09440-t001:** Characteristics of the participants (*n* = 101).

Variable		Mean	SD
Age		44.0	9.2
Onset		23.3	6.5
Psychiatric history in years		20.7	9.2
Education in years		9.4	1.9
C-MMSE		29.9	2.5
**Variable**		**N**	**%**
Gender (Male/Female)		60/41	59.4/40.6
CGI-S	Not at all	11	10.9
	Mild	52	51.5
	Borderline	38	37.6

Note: SD = standard deviation; CGI-S = Clinical Global Impression–Severity; C-MMSE = Chinese version of the Mini-Mental State Examination.

**Table 2 ijerph-18-09440-t002:** Test–retest reliability of the RSAT (raw scores).

Measure	1st Test M (SD)	2nd Test M (SD)	Difference M (SD)	*p*-Values	Effect Size	ICC (95% CI)	SEM	MDC (MDC %)	Heteroscedasticity (Pearson *r*)
ADS	143.3 (49.7)	150.3 (53.6)	7.03 (21.3)	0.001 *	−0.14	0.91 (0.86–0.94)	14.9	41.3 (28.1)	0.38
ADE	5.8 (7.4)	4.5 (6.1)	−1.4 (5.1)	0.008 *	0.18	0.70 (0.59–0.79)	4.1	11.3 (218.9)	0.58
ADA	95.8 (5.2)	97.1 (4.5)	1.2 (3.7)	0.001 *	−0.24	0.69 (0.56–0.79)	2.9	8.0(8.3)	−0.58
CSS	118.3 (38.0)	129.7 (40.3)	11.4 (20.1)	0.000 *	−0.30	0.83 (0.66–0.90)	15.7	43.4 (35.0)	0.28
CSE	10.8 (9.2)	8.8 (7.8)	−2.1 (6.0)	0.001 *	0.23	0.73 (0.60–0.82)	4.8	13.3 (135.5)	0.41
CSA	91.3 (8.5)	93.4 (7.0)	2.1 (4.7)	0.000 *	−0.25	0.79 (0.65–0.87)	3.9	10.8 (11.6)	0.31

Notes: ADS = Automatic Detection Speed; ADE = Automatic Detection Errors; ADA = Automatic Detection Accuracy; CSS = Controlled Search Speed; CSE = Controlled Search Errors; CSA = Controlled Search Accuracy; * = *p*-values < 0.05.

**Table 3 ijerph-18-09440-t003:** Comparison of the proposed results with those of related studies.

Author	Year	Participants	Task	Results
Lemay et al. [62]	2004	Middle-aged to elderly	RSAT	Speed ICC: 0.82 Accuracy ICC: 0.68
Messinis et al. [31]	2007	Greek adults	RSAT	Speed ICC: 0.94–0.98 Accuracy ICC: 0.73–0.89
Knight et al. [34]	2010	Older adults	RSAT	Speed SEM: 7.46–7.93 Speed practice effect: 3.4–6.1 Accuracy SEM: 3.00–3.65 Accuracy practice effect: 0.1–0.3
Tang et al. [25]	2018	Patients with schizophrenia	SDMT	ICC: 0.91–0.94 SEM: 3.0–3.8 Effect sizes: 0.02–0.35
			T-SDMT	ICC: 0.89–0.94 SEM: 2.5–3.3 Effect sizes: 0.07–0.43
Muliady et al. [66]	2019	Patients with schizophrenia	BACS-I	ICC: 0.94
Chen et al. [67]	2020	Patients with schizophrenia	CPT-IP	ICC: 0.62–0.88 MDC%: 33.8–110.8 Effect sizes: −0.13–0.24
Shih et al. [68]	2021	Patients with schizophrenia	SHTA	ICC: 0.67 MDC%: 12.1
Chiu et al. [69]	2021	Patients with schizophrenia	WCST	ICC: 0.7 MDC: 3.3–42.0 Effect sizes: 0.03–0.13
Chen et al. [70]	2021	Patients with schizophrenia	TONI-4	ICC: 0.73 MDC: 5.1, MDC%: 14.2 Effect sizes: −0.03
The Presented Methods	2021	Patients with schizophrenia	RSAT	ICC: 0.69–0.91 MDC: 8.0–43.4, MDC%: 8.3–218.9 Effect sizes: −0.14–0.30

Notes: RSAT = Ruff 2 and 7 Selective Attention Test; SDMT = Symbol Digit Modalities Test; T-SDMT = Tablet-based Symbol Digit Modalities Test; BACS-I = Indonesian version of the Brief Assessment of Cognition in Schizophrenia; CPT-IP = Continuous Performance Test, Identical Pairs version; SHTA = Shih–Hsu Test of Attention; WCST = Wisconsin Card Sorting Test; TONI-4 = Test of Nonverbal Intelligence–Fourth Edition.

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
