# Peer review of "Practice Effects, Test–Retest Reliability, and Minimal Detectable Change of the Ruff 2 and 7 Selective Attention Test in Patients with Schizophrenia"

_ijerph, 2021, doi:10.3390/ijerph18189440_

Round 1

Reviewer 1 Report

The manuscript "Practice Effects, Test–Retest Reliability, and Minimal Detecta-2 ble Change of the Ruff 2 and 7 Selective Attention Test in Patients with Schizophrenia" presents an study about test-retest reliability, effect, and minimum detectable change. In general the manuscript is well written, and methodologically correct.

To know MDC, practice effects and test-retest variability is very important, and to have that information for subjects with Schizophrenic condition is very useful. In terms of the format, some figures are low resolutions and need to be improved (please use vectored format when possible).

Author Response

Responses to Reviewers

Manuscript ID:  ijerph-1335124

Manuscript Title: Practice Effects, Test–Retest Reliability, and Minimal Detectable Change of the Ruff 2 and 7 Selective Attention Test in Patients with Schizophrenia

We would like to thank you for your valuable comments, which have greatly helped us to improve the manuscript. We have carefully considered the comments and substantially improved the manuscript. Each revision and comment suggested by the reviewers has been considered and accurately incorporated. We have carefully addressed and responded to the Reviewers’ valuable comments and suggestions in the revised manuscript.

All modifications have been made in the revised manuscript and are highlighted in yellow.

We herein provide point-by-point responses to the reviewers’ comments with references to the changes made in the text. For your convenience, we also provide a point-by-point list of actions with reference to the changes made in the text.

Reviewer 1

Comment 1: The manuscript "Practice Effects, Test–Retest Reliability, and Minimal Detectable Change of the Ruff 2 and 7 Selective Attention Test in Patients with Schizophrenia" presents a study about test-retest reliability, effect, and minimum detectable change. In general the manuscript is well written, and methodologically correct.

Response: Thank you very much for the positive recommendation and comments.

Comment 2: To know MDC, practice effects and test-retest variability is very important, and to have that information for subjects with Schizophrenic condition is very useful. In terms of the format, some figures are low resolutions and need to be improved (please use vectored format when possible).

Response: Thank you for the kind suggestions. In the revised manuscript, we have reprocessed the figures at a higher resolution. The revised Figure 2 is shown below.

Reviewer 2 Report

General Comment:

This paper introduces a study to examine the test-reset reliability, practice effect and minimal detectable change in patients with schizophrenia using the Ruff 2 and 7 Selective Attention Test (RSAT). The major contribution of the work is the deep assessment of the RSAT in both qualitative and quantitative fashion. Performance results are presented by difference scores between two scenarios, which gives valuable result for the proposed method. The paper is well written, has a correct structure and is mostly clear. I feel that the paper has value to be published. Some suggestions are proposed to improve the paper readability and quality, though.

Minor comments:

  • Line 209: should be following equations
  • Authors should explicit the meaning of the x and y axis of Figure 2. At a glance, it is not trivial to deduce the menaing of dADS, mADS, etc…

Author Response

Responses to Reviewers

Manuscript ID:  ijerph-1335124

Manuscript Title: Practice Effects, Test–Retest Reliability, and Minimal Detectable Change of the Ruff 2 and 7 Selective Attention Test in Patients with Schizophrenia

We would like to thank you for your valuable comments, which have greatly helped us to improve the manuscript. We have carefully considered the comments and substantially improved the manuscript. Each revision and comment suggested by the reviewers has been considered and accurately incorporated. We have carefully addressed and responded to the Reviewers’ valuable comments and suggestions in the revised manuscript.

All modifications have been made in the revised manuscript and are highlighted in yellow.

We herein provide point-by-point responses to the reviewers’ comments with references to the changes made in the text. For your convenience, we also provide a point-by-point list of actions with reference to the changes made in the text.

Reviewer 2

Comment 1: This paper introduces a study to examine the test-reset reliability, practice effect and minimal detectable change in patients with schizophrenia using the Ruff 2 and 7 Selective Attention Test (RSAT). The major contribution of the work is the deep assessment of the RSAT in both qualitative and quantitative fashion. Performance results are presented by difference scores between two scenarios, which gives valuable result for the proposed method. The paper is well written, has a correct structure and is mostly clear. I feel that the paper has value to be published. Some suggestions are proposed to improve the paper readability and quality, though.

Response: Thank you very much for the positive recommendation and comments. We have carefully addressed and responded to the Reviewer’s valuable comments and suggestions in the revised manuscript.

Comment 2: Line 209: should be following equations

Response: Thank you for the suggestion. We have moved equation (1) to Line 209. The revised sentence is as follows:

“The minimal detectable change (MDC) was calculated based on the ICC value with the following equations (1) and (2): SEM = SD first session of testing scores × (  )             (1)”

Comment 3: Authors should explicit the meaning of the x and y axis of Figure 2. At a glance, it is not trivial to deduce the meaning of dADS, mADS, etc…

Response: Thank you for the kind suggestion. In the revised manuscript, we have reprocessed Figure 2 at a higher resolution and added explicit labels for the x and y axes to each sub-figure of Figure 2. The meaning of the x-axis is "mean of test–retest of ...", so mADS has been modified to “Mean of Test–retest of ADS”. The meaning of the y-axis is "difference between test–retest of ...", so dADS has been modified to “Difference Between Test–retest of ADS”. The same applies to the other sub-figures. The revised Figure 2 is shown below.

Reviewer 3 Report

The topic of this study is very interesting and important; however I feel that this manuscript it is not ready for publication.

Please see the attached file with several comments.

Author Response

Responses to Reviewers

Manuscript ID:  ijerph-1335124

Manuscript Title: Practice Effects, Test–Retest Reliability, and Minimal Detectable Change of the Ruff 2 and 7 Selective Attention Test in Patients with Schizophrenia

We would like to thank you for your valuable comments, which have greatly helped us to improve the manuscript. We have carefully considered the comments and substantially improved the manuscript. Each revision and comment suggested by the reviewers has been considered and accurately incorporated. We have carefully addressed and responded to the Reviewers’ valuable comments and suggestions in the revised manuscript.

All modifications have been made in the revised manuscript and are highlighted in yellow.

We herein provide point-by-point responses to the reviewers’ comments with references to the changes made in the text. For your convenience, we also provide a point-by-point list of actions with reference to the changes made in the text.

Reviewer 3

Comment 1: The topic of this study is very interesting and important; however, I feel that this manuscript it is not ready for publication.

Response: Thank you for the comment. We have carefully considered the comments, which helped us to improve the manuscript substantially.

Comment 2: In Line 84-90, please reword the sentence in order of start differently.

Response: Thank you for the suggestion. In this revision, we have reworded the sentence as follows:

“One index of random measurement error that can be used to present the precision of individual results is SEM. SEM is estimated from the standard deviation of a sample of scores at baseline and a test–retest reliability index of the measurement instrument. The minimal threshold beyond random measurement error at certain confidence levels between two assessments is called MDC. MDC is estimated from SEM and a degree of confidence [27].”

Comment 3: In Line 93-94, please reword the sentence, starts the same way several times.

Response: Thank you for the suggestion on style. In this revision, we have reworded the sentence as

“The MDC can further be calculated as the MDC percentage (MDC%), which can be used to identify random measurement error.”

Comment 4: In Line 97, please add the reference of the original author.

Response: Thank you for pointing out our omission. We have added a reference to the original author after the subsubsection title. In this revision, the title of the subsubsection is “Ruff 2 and 7 Selective Attention Test (RSAT) [29]”

Comment 5: In Line 156, please add information regarding the time it takes the data collection of each participant. The data collection was made individually or in group?

Response: Thank you for the suggestion. All participants individually received one-on-one RSAT assessments by the same assessor. A single assessment includes instructions, practice and a formal exam. The total assessment time is about 15 minutes. In this revision, we have added an explanation of the data collection as follows:

"All participants individually received one-on-one RSAT assessments by the same assessor. Such an assessment includes instructions, practice, and a formal exam. The total assessment time is about 15 minutes.”

Comment 6: In Line 159, please add information regarding which version was used and give psychometric information regarding this version.

Response: Thank you for the helpful suggestion. We used the version compiled by the original author [29]. The test–retest reliability has been reported as adequate to high, with higher test–retest coefficients reported for Speed than for Accuracy scores. In this revision, we have added information about which version was used and provide psychometric information regarding that version as follows:

“This study used the version compiled by the original author, for which the alpha and split-half coefficients for the normative sample are high, suggesting good internal reliability.”

Comment 7: In Line 200, please remove the point.

Response: Thank you for the kind reminder. We have deleted the period between the sentence and in-text citation. The revised sentence is as follows:

“ICC values < 0.40, between 0.40 and 0.59, between 0.60 and 0.79, and ≥ 0.80 respectively indicated poor, fair, good, and excellent reproducibility [50–53].”

Comment 8: In Line 246-248, SD.

Response: Thank you for your kind reminder. We have deleted the SD and values of SD in the text and relegated that information to Table 1. The revised sentence are as follows:

“Table 1 shows the characteristics of the participants. The mean age was 44.0 years, and 59.4% of the participants were male. The mean onset age was 23.3 years. The mean duration of psychiatric history was 20.7 years.”

Comment 9: In Line 315-317, please add references.

Response: Thank you for pointing out our omission. We have added a reference. The revised sentences are as follows:

“A linear performance increase has also been observed in various tests of attention. Younger adults also demonstrate larger practice effects than those of older participants [34].”

Comment 10: In Line 355, please add this sentence to the previous paragraph.

Response: Thank you for the useful suggestion. We have moved this sentence to the previous paragraph. The revised paragraphs are as follows:

“Accordingly, a fixed value of MDC is not appropriate for patients with diverse levels of performance. The second reason is that the RSAT has a trivial-to-small practice effect, indicating that the scores systematically increase in repeated assessments.

    Thus, patients’ change scores are more likely to exceed the MDC value and thus lead to overestimation.”

Comment 11: In Line 380, tables should appear in results section.

Response: Table 3 is not a summary of the results of this study but a discussion and comparison with similar studies, which is why it is presented in the discussion section.

Comment 12: In Line 398, please add practical implications of the study; proposals for future studies.

Response: Thank you for the suggestion. We have added practical implications of the study and proposals for future studies. The added sentences are as follows:

“The practical implications of the study are that the RSAT is recommended for clinical and research applications because it is reliable in patients with schizophrenia. Our sample was a convenience sample of inpatients. In future studies, it will be necessary to expand the scope of sampling.”
